# An Efficient and Robust Deep Learning Method with 1-D Octave Convolution to Extract Fetal Electrocardiogram

**DOI:** 10.3390/s20133757

**Published:** 2020-07-04

**Authors:** Khuong Vo, Tai Le, Amir M. Rahmani, Nikil Dutt, Hung Cao

**Affiliations:** 1Donald Bren School of Information and Computer Sciences, University of California, Irvine, CA 92697, USA; khuongav@uci.edu (K.V.); a.rahmani@uci.edu (A.M.R.); 2Henry Samueli School of Engineering, University of California, Irvine, CA 92697, USA; tail3@uci.edu; 3Sue & Bill Gross School of Nursing, University of California, Irvine, CA 92697, USA

**Keywords:** non-invasive fetal electrocardiogram, QRS complexes, deep learning, neural networks, octave convolution

## Abstract

The invasive method of fetal electrocardiogram (fECG) monitoring is widely used with electrodes directly attached to the fetal scalp. There are potential risks such as infection and, thus, it is usually carried out during labor in rare cases. Recent advances in electronics and technologies have enabled fECG monitoring from the early stages of pregnancy through fECG extraction from the combined fetal/maternal ECG (f/mECG) signal recorded non-invasively in the abdominal area of the mother. However, cumbersome algorithms that require the reference maternal ECG as well as heavy feature crafting makes out-of-clinics fECG monitoring in daily life not yet feasible. To address these challenges, we proposed a pure end-to-end deep learning model to detect fetal QRS complexes (i.e., the main spikes observed on a fetal ECG waveform). Additionally, the model has the residual network (ResNet) architecture that adopts the novel 1-D octave convolution (OctConv) for learning multiple temporal frequency features, which in turn reduce memory and computational cost. Importantly, the model is capable of highlighting the contribution of regions that are more prominent for the detection. To evaluate our approach, data from the PhysioNet 2013 Challenge with labeled QRS complex annotations were used in the original form, and the data were then modified with Gaussian and motion noise, mimicking real-world scenarios. The model can achieve a F_1_ score of 91.1% while being able to save more than 50% computing cost with less than 2% performance degradation, demonstrating the effectiveness of our method.

## 1. Introduction

The U.S. fetal mortality rate has remained unchanged from 2006 through 2012 at 6.05 per 1000 births [1], motivating the need for proactive fetal monitoring techniques to alert and reduce fetal mortality. The COVID-19 pandemic has revealed the weaknesses of our healthcare system in providing remote monitoring for essential services. Although mobile health and telemedicine technologies have been introduced for more than a decade, expectant mothers still need to visit clinics in person for regular checkups, especially twice or thrice a week in the last month of pregnancy. Additional technological improvements can enable these fetal well-being monitoring and non-stress tests to be performed remotely, which can result in significant cost and time reduction, as well as mitigating the burden for the hospitals, while supporting social distancing when needed.

Continuous fetal heart rate monitoring, i.e., cardiotocography (CTG), is associated with a decrease in fetal mortality [2], with Doppler ultrasound-based fetal heart rate (fHR) monitoring being the most popular. However, this measurement technique is especially challenging for persons without medical knowledge and experience, and the Food and Drug Administration (FDA) has also warned about its potential risk [3]. Alternatively, fetal heart monitoring can also be done via fetal electrocardiogram (fECG) acquired from the fetal scalp, but this is an invasive procedure and can potentially cause some risks such as infection. Non-invasive fECG is a promising technique that does not come with those limitations. It can be achieved by measuring through the mother’s abdominal signal that includes both the maternal ECG (mECG) and fECG. Such measures can help diagnose problems such as preterm delivery, hypoxia, intrauterine growth retardation, etc., which may not only pose a risk to the fetus, but also to maternal health [4]. Furthermore, monitoring and analysis of fECG can also provide information to enable early diagnosis of congenital heart disease (CHD), which was estimated to affect 1 million children in the U.S. in 2010 [5]. Therefore, the extraction of fECG from the abdominal signal is of utmost importance.

Blind source separation (BSS) methods, template subtraction (TS), and filtering techniques are popular approaches used to extract fECG [6]. The BSS methods assume that the abdominal signal is a combination of independent sources, including fECG, mECG, and noises [7]. By finding a matrix through the process of maximizing the statistical independence of each source in the abdominal signal, the extracted signal was estimated. This, however, is computationally time consuming, thereby, not ideal for BSS implementation on mobile devices. While the TS method has the advantage of simplicity, the overlap between the fECG and mECG waveforms causes the wrong detection of fetus’ ECG peaks. Moreover, finding the best mECG template for subtraction is also challenging as it requires a reliable mECG reference and it sometimes does not correctly represent mECG signal collected from the mother’s abdomen. Filtering methods [8,9] mostly based on variants of the adaptive filtering technique either remove the mECG using other maternal reference signals or directly extract the fECG by canceling the mECG. Since this requires an additional reference signal correlated to mECG, it makes this method more complicated.

Recently, deep learning methods based on convolutional neural networks (CNNs) were introduced to perform fECG extraction. CNNs detect fetal QRS (fQRS) complexes based only on non-invasive fetal ECG. Zhong et al. [10] designed a 2-D CNN model with three convolutional layers to extract features from single-channel fetal ECG signals, while the model developed by Lee et al. [11] has a deeper architecture and uses multi-channel signals, which lead to the performance improvement. Besides, the time-frequency representation of the abdominal ECG recordings is also applied to feed to a CNN model [12]. However, the performance of these methods depends heavily on pre- or post-processing procedures. Additionally, vanilla CNNs cannot run efficiently on resource-constrained devices (e.g., wearables) due to their high computational intensity, which poses a critical challenge for real-world deployment.

In this paper, we propose an efficient end-to-end deep neural network that accurately detects fQRS complexes from multi-channel non-invasive fECG, without requiring a reference maternal ECG signal. The contributions of this work are three-fold: (1) We extended the 2-D octave convolution (OctConv) [13] to 1-D OctConv for factorizing hierarchical mixed-frequency f/mECG features at multiple network layers. This is motivated by the observation shown in Figure 1, where a f/mECG signal can be decomposed into high- and low-frequency components, in which the low-frequency part focuses on global structure or low variations of the signal, while the high-frequency part describes details or high varying parts. OctConv inherently captures more contextual information at each layer while reducing temporal redundancy, which leads to potential performance improvement at considerably less memory and computational resources. To the best of our knowledge, this is the first study to show the effectiveness of 1-D octave convolution in time-series biomedical signal. (2) Our method requires no significant data pre-processing or feature engineering, while achieving higher or comparable performance to the state-of-the-art methods. The proposed model is a combination of convolutional and recurrent neural networks, with a residual network (ResNet) architecture inspired by the network of Wang et al. [14], which was proven to achieve superior classification performance on a variety of time-series datasets [15]. (3) In order to minimize the risks of using black-box models and better adapt to clinical settings, we adapted the gradient-weighted class activation mapping (Grad-CAM) method [16] from 2-D images to 1-D time series to generate class activation maps. This supports clinicians in interpreting and understanding the model’s decision.

The rest of our paper is organized as follows. Section 2 presents the experimental data and our proposed architecture, as well as provides the efficiency analysis. In Section 3, experimental results are discussed along with the interpretation of the model’s predictions. Finally, Section 4 concludes the paper.

## 2. Materials and Methods

### 2.1. Experimental Data

In this research, set A from the PhysioNet/computing in the cardiology challenge database (PCDB) [17] was used as the experimental. This is the largest publicly available, non-invasive fECG database to date, which consists of a collection of one-minute abdominal ECG recordings. Each recording includes four non-invasive abdominal channels where each channel is acquired at 1000 Hz. The data were obtained from multiple sources using a variety of instrumentation with differing frequency response, resolution, and configuration. We used the set A, containing 75 records with reference annotations, excluding a number of recordings (a33, a38, a47, a52, a54, a71, and a74) having inaccurate reference annotations [18]. The annotations of the locations of fetal QRS complexes were manually produced by a team of experts, usually with reference to direct fECG signals that were acquired from a fetal scalp electrode [17]. As suggested in [10], the validation set and test set were comprised of six recordings (a08–a13) and seven recordings (a01–a07), respectively. The 55 remaining recordings were for the training set.

To evaluate the effectiveness of our method in practical scenarios, Gaussian noise was added with various noise levels, making the signal-to-noise ratio (SNR) of the modified dataset with 50.6 dB, 36.8 dB, and 29.1 dB, respectively. Specifically, a normally distributed random noise ranging from −4 μV to 4 μV was generated by *randn* function in MATLAB (R2018b, the Mathworks, Inc. Natick, MA, USA). To have different noise levels (i.e., different SNR values), we multiplied the initial random noise with constant numbers (e.g., 3, 6, and 9) before adding to the original dataset. Additionally, since the dataset was collected in a clinical setting where motion artifacts were mostly avoided because the subjects were in resting positions, motion artifacts were added to the data. For this purpose, the ECG data were recorded from a healthy subject in different types of activities such as walking and jogging. The acquired data were normalized between [−1, 1] and motion noise and filtered ECG data were then achieved by using extended Kalman filter. The f/mECG was normalized with the same threshold of [−1, 1] before motion noise was added to the normalized f/mECG.

In the context of fetal ECG extraction, a detected fetal QRS is considered as a true positive if it is within 50 ms of the reference annotation. Therefore, for every input window frame of 4 × 100, it is labeled as Class 1 if it contained the location of the complex, otherwise labeled as Class 0.

### 2.2. Model Architecture

As shown in Figure 2, a fully convolutional network with shortcut residual connections is applied to the input data size 4 × 1000 of a 1-second window of time, followed by a recurrent network layer to output 10 predictions over 10 consecutive window frames of 4 × 100. A general 1-D convolution is applied for a centered time stamp, *t*, by the following equation:
(1)$$Y_{t}=f\left(\omega \;*\;X_{t-l/2:t+l/2}+b\right)\;|\;\forall t\in \left[1,\;T\right]$$
where *Y* denotes the resulting feature maps from a dot product of time series *X* of length *T* with a filter *ω* of length *l*, a bias parameter *b*, and a non-linear function *f*. As can be seen in Figure 3a, all input and output feature maps in vanilla convolution retains the same temporal resolution across the channels, which possibly introduces temporal redundancy since low-frequency information is captured by some feature maps that can be further compressed.

With this observation, we adapted the 2-D octave convolution (OctConv) [13] to the multi-channel f/mECG with 1-D OctConv, as illustrated in Figure 3b. Input feature maps of a convolutional layer X can be factorized along the channel dimension into low-frequency maps $X^{L}$ and high-frequency maps $X^{H}$. High-frequency channels retains the feature map’s resolution, while low-frequency channels down-sampled it. Practically, the $X^{L}$ feature representation has the temporal dimension or the length divided by 2, an *octave*, to produce two frequency blocks. There is a hyper-parameter α ϵ [0, 1] to determine the ratio of channels allocated to the low-resolution features. The octave convolution effectively processes feature maps directly in their corresponding frequency tensors and manages the information exchange between frequencies, by having different filters focusing on different frequencies of the signal. For convolving on $X^{H}$ and $X^{L}$, the original convolutional kernel W is split into two components, $W^{H}$ and $W^{L}$, which are further broken down to intra- and inter-frequency parts: $W^{H}=[W^{H\to H},W^{H\to L}]$ and $W^{L}=[W^{L\to L},W^{L\to H}]$. Let $Y^{H}$ and $Y^{L}$ represent the high-frequency and low-frequency output tensors, octave convolution can be written as:(2)$$\begin{array}{l} Y^{H}=f(X^{H};W^{H\to H})+upsample(f(X^{L};W^{L\to H}),2)\\ Y^{L}=f(X^{L};W^{L\to L})+f(pool(X^{H},2);W^{H\to L}) \end{array}$$
where upsample(X;k) is an up-sampling operation by a factor of k via nearest interpolation, pool(X,k) is an average pooling operation with kernel size k and stride k, and f(X;W) denotes a convolution with parameters W.

The benefit of the new feature representation and convolution operation is the reduction in computational cost and memory footprint, enabling the implementation on resource-constrained devices. This also helps convolutional layers capture more contextual information by having larger receptive fields, by having low-frequency part $X^{L}$ convolved by 1×k kernels, which could lead to the improvement in the classification performance.

The 1-D CNN comprises of nine convolutional layers and one global average pooling layer. The nine convolutional layers are divided into three residual blocks with the shortcut residual connections between three consecutive blocks. Those connections are linear operations that link a residual block’s output to its input to alleviate the vanishing gradient effect by allowing the gradient to propagate directly through these links. In each residual block, the first, second, and third convolution have the filter lengths of 8, 5, and 3, respectively. The total numbers of filters are 64, 128, and 128 for the first, second, and third block, respectively. Each block is followed by the rectified linear unit (ReLU) activation preceded by a batch normalization (BN) operation [19]. The length of the input time series is not altered after convolutions with strides of 1 and appropriate padding. This ResNet architecture can be made invariant across different time-series datasets [15], making it suitable for transfer learning techniques [20] in which the model is first trained on source datasets and then fine-tuned on the target dataset to further improve performance.

After the convolutional layers, we have the feature representation of the size 128 × 1000. This feature is further split into 10 parts of 128 × 100, and then each part is global-average pooled across the time dimension resulting in a 128-dimensional vector. The 10 × 128 tensor of 10 timesteps is then fed to the recurrent network layer of gated recurrent units (GRUs) [21] with the hidden state size of 32 to capture the sequential nature of QRS complexes. GRUs have gating mechanisms that adjust the information flow inside the unit. A GRU memory cell is updated at every timestep, *t*, as detailed in the following equations:(3)$$\begin{array}{l} r_{t}=\sigma (W_{r}x_{t}+U_{r}h_{{}_{t-1}})\\ z_{t}=\sigma (W_{z}x_{t}+U_{z}h_{t-1})\\ {\hat{h}}_{t}=\tanh(Wx_{t}+U(r_{t}⊙h_{t-1}))\\ h_{t}=(1-z_{t})h_{t-1}+z_{t}{\hat{h}}_{t} \end{array}$$
where *σ* denotes the logistic sigmoid function, ⊙ is the elementwise multiplication, *r* is the reset gate, *z* is update gate, and ${\hat{h}}_{t}$ denotes the candidate hidden layer.

In addition, since GRUs only focus on learning dependencies in one direction with the assumption that the output at timestep t only depends on previous timesteps, we deploy bi-directional GRUs to capture dependencies in both directions. Finally, the shared-weight softmax layer was applied in each feature region corresponding to each of the 4 × 100 input data to detect the fQRS complexes. Note that the recurrent layer and the softmax layer on top yield negligible computing cost by modeling only a short sequence of 10 timesteps with a minimal hidden state size.

### 2.3. Theoretical Gains of 1-D OctConv

Memory cost. For feature representation with vanilla 1-D convolution, the storage cost is l×c, where *l* and *c* are the length and the number of channels, respectively. In OctConv, a low-frequency tensor is stored at 2× lower temporal resolution in α×c channels, as illustrated in Figure 3, thus storage cost of the multi-frequency feature representation is $l\times (1-\alpha )\times c+\frac{l}{2}\times \alpha \times c$. Therefore, the memory cost ratio between the 1-D OctConv and regular convolution is
(4)$$1-\frac{1}{2}\times \alpha .$$

Let k be the filter length, the floating point operations per second (FLOPs) (i.e., multiplications and additions) for computing output feature map in vanilla 1-D convolution can be calculated by
(5)$$FLOPs(f_{Conv})=l\times k\times c^{2}.$$

In OctConv, the main computation comes from the convolution operations in the two paths of inter-frequency information exchange ($f^{H\to L}$ and $f^{L\to H}$), and the two paths of intra-frequency information update ($f^{H\to H}$ and $f^{L\to L}$). They are estimated for each path as below.
(6)$$\begin{array}{l} FLOPs(f^{H\to H})=l\times k\times {\left(1-\alpha \right)}^{2}\times c^{2}\\ FLOPs(f^{H\to L})=\frac{l}{2}\times k\times \alpha \times (1-\alpha )\times c^{2}\\ FLOPs(f^{L\to H})=\frac{l}{2}\times k\times (1-\alpha )\times \alpha \times c^{2}\\ FLOPs(f^{L\to L})=\frac{l}{2}\times k\times \alpha^{2}\times c^{2} \end{array}$$

Hence, the total cost for computing output feature map with OctConv is
(7)$$FLOPs(f_{OctConv})=l\times k\times c^{2}\times (1-\alpha +\frac{\alpha^{2}}{2}).$$

Therefore, the computational cost ratio between the 1-D OctConv and regular convolution is
(8)$$1-\alpha \times (1-\frac{\alpha }{2}).$$

From Equations (4) and (8), we can derive the theoretical gains of the 1-D octave convolution per layer regarding different values of α, as shown in Table 1.

### 2.4. Visualization of Class Discriminative Regions

As a deep learning method, the model is inherently a black-box function approximator, which is one of the primary obstacles for its use in medical diagnosis. In order to make the model more transparent, we extended the gradient-weighted class activation mapping (Grad-CAM) technique [16] to highlight the discriminative time-series regions that contribute to the final classification. As opposed to the model-agnostic interpretation methods [22,23], which require many forward passes, Grad-CAM has a fast-computation advantage by requiring only one single pass. In contrast to the CAM method [24,25] that trades off model complexity and performance for more transparency, Grad-CAM has the flexibility in applying to any CNN-based models in which CNN layers could be followed by recurrent neural network layers or fully connected layers. The method relies on the linear combination of the channels in the output feature maps of the last convolutional layers, which is the 128-channel feature map that appears before the global-average pooling layer in our network. Grad-CAM computes to the weights $a_{k}^{c}$, which is the importance of channel k for the target class c, by averaging gradients across the time dimension of the score for Class c$\left(y^{c}\right)$ with respect to channels A:(9)$$\alpha_{k}^{c}=\frac{1}{m}\sum_{j}\frac{\partial y^{c}}{\partial A_{j}^{k}}.$$

Since only features that have positive influences on the target class are important, ReLu is applied to the linear combination of maps to derive the heat map for the prediction of Class c:(10)$$L_{Grad-CAM}^{c}=ReLu(\sum_{k}\alpha_{k}^{c}A^{k}).$$

## 3. Results and Discussion

### 3.1. Experiment Setup

Model parameters θ are achieved by minimizing the weighted binary cross entropy loss between the ground truth targets or labels y and the estimates or predictions $\hat{y}$. That is,
(11)$$\hat{\theta }={\mathrm{argmin}}_{\theta }-\frac{1}{M}\overset{M}{\underset{i=1}{\sum }}\overset{T}{\underset{j=1}{\sum }}(\beta y_{ij}\log{\hat{y}}_{ij}-\left(1-y_{ij}\right)\log\left(1-{\hat{y}}_{ij}\right))$$
where M denotes the number of training samples, T is equal to 10, which is the number of successive sub-sequences to detect fQRS complexes, $y_{ij}$ signifies the complex at sequence j of ith training sample, and the multiplicative coefficient *β* is set to 2, which up-weights the cost of a positive error relative to a negative error in order to alleviate the imbalanced class problem between the number of fQRS complexes and non-fQRS complexes. The model was trained with Adam optimizer [26] with the initial learning rate of 0.001, the exponential decay rates β1 = 0.9 and β2 = 0.999, and the constant for numerical stability ϵ = 1e–8. All trainable weights were initialized with Xavier/Glorot initialization scheme [27]. All dropout rates [28] were set at 0.4 to prevent the neural network from overfitting. The training process ran for a total of 100 epochs with batch size of 64, and the best model on the validation set was chosen to report its performance on the test set.

### 3.2. Evaluation Metrics

The fundamental measures for evaluating classification performance are precision, recall, and F_1_ score. Precision is a measure of exactness that depicts the capacity of the model at detecting true fQRS complexes out of all the detections it makes. Recall is a measure of completeness that depicts the model’s capacity at finding the true fQRS complexes. F_1_ score is the harmonic mean of precision and recall. They are calculated as
(12)$$\mathrm{Precision}=\frac{\mathrm{TP}}{\mathrm{TP}+\mathrm{FP}}$$
(13)$$\mathrm{Recall}=\frac{\mathrm{TP}}{\mathrm{TP}+\mathrm{FN}}$$
(14)$$F_{1}=\frac{2\times \mathrm{Precison}\times \mathrm{Recall}}{\mathrm{Precision}+\mathrm{Recall}}$$
where TP is the number of true positives (correctly identified fQRS), FP is the number of false positives (wrongly detected fQRS), and FN is the number of false negatives (missed fQRS).

### 3.3. Results and Interpretations

In Table 2, we measure the performance of the proposed model with varying α ϵ [0, 0.25, 0.5, 0.75] on the original PhysioNet dataset. The constant number of model parameters was 0.56 million. At α = 0, the model with the vanilla convolution had the computational cost of 0.52 gigaFLOPs (GFLOPs). The computation of the recurrent layer and the softmax layer (GRU-FC-GFLOPs) was fixed at approximately 0.0003 GFLOPs, which contributed only 0.06% of the total cost. We observed that the F_1_ score on the test dataset (F_1_-test) first increased marginally and then slowly declined with the growth of α. The highest F_1_ score of 0.911 was reached at α = 0.25 when the computation of the convolutional layers (CNN-GFLOPs) was reduced by around 20%. We attributed the increase in F_1_ score to OctConv’s effective design of multi-frequency processing and the resultant contextual-information augmentation by the enlargement of receptive fields. It is interesting to note that the compression to half the resolution of 75% feature maps resulted in only 1.4% F_1_ score drop. To better support the generalizability of our approach, we also performed cross-validation to obtain means and standard deviations of the model’s performance over 10 folds of the dataset (F_1_-cross). We observed that the cross-validated performance showed a similar trend with the results on the selected test set, with even smaller F_1_ score gaps between different α. Likewise, the GPU inference time also diminished with the drop in the number of FLOPs and the increase in α. These results demonstrated 1-D OctConv’s capability of grouping and compressing the smoothly changed time-series feature maps. Note that OctConv is orthogonal and complementary to existing methods for improving CNNs’ efficiency. By combining OctConv with popular techniques, such as pruning [29] and depth-wise convolutions [30], we can further cut down the computing cost of the model.

Figure 4 depicts the effect of the number of timesteps fed into the recurrent layer with GRUs. At each timestep, the model processed the feature corresponding to the input window frame of 4 × 100. The F_1_-score grew sharply with the increase in the sequence length and reached the peak at 10 timesteps before leveling off. Also, it is worth noting that the computational cost and memory footprint of the model grow with the increase in the input sequence length. This result validated our input segmentation strategy as well as showed the necessity of the recurrent layer to model the sequential nature of the fQRS complexes.

We also compared our approach with the two recent deep learning algorithms reported in the literature [9,10] in Figure 5. Our method had a higher recall (90.32%) than those of other algorithms (89.06% and 80.54%), while the precision (91.82%) was slightly lower than Lee’s algorithm (92.77%) but significantly higher than Zhong’s approach (75.33%). In Lee’s model, the architecture is manually tuned as well as requiring post-processing steps to be best optimized for the specific task, while our proposed method performed only a single stage of processing and can generalize easily to other signal-processing schemes by having the ResNet architecture invariant across various time-series classification tasks as proven in [15].

Table 3 shows the model performance on the added-noise datasets in order to evaluate the effectiveness of our method in real-life scenarios. Regarding Gaussian noise, the performance fell sharply with the increase in the noise level. The F_1_ score was 0.815 at noise level 3 but decreased to 0.627 at noise level 9 when data was completely corrupted. Besides, the model achieved the F_1_ score of 0.844 on the dataset disturbed with motion artifacts. These promising results demonstrate the robustness of our method in practical scenarios against different types of noise.

Figure 6 shows our attempt to interpret the inner workings of our network. In Figure 6a, we show the examples of the heatmap outputs in a window frame using the Grad-CAM approach. The technique enabled the classification-trained model to not only answer whether a time series contains the location of the complex, but also localize class-specific regions. This provided us with the confidence that the detection of the QRS complexes inside a window frame is because the model correctly focused on the highest discriminative subsequence around R-peak (i.e., the maximum amplitude in the R wave of the ECG signal), and not for unknown reasons. Figure 6b,c illustrate the attention mappings for the detections of one of the fQRS complexes by our model on the entire input signals. There were two fQRS complexes and three complexes detected associated with the attention spikes in Figure 6b,c, respectively. Although the maternal ECG signal is the predominant interference source, possessing much greater amplitude than the fetal ECG, our network still managed to pay high attention to the right regions of the waveform that contain fetal QRS complexes. Moreover, when deciding on each window frame, the model not only focused on the local morphology of fQRS complexes, but it also took into account the surrounding complexes to reinforce its decision, which proved the capability of the recurrent layer in our network. Besides the high accuracy, this demonstrates that our model understands the problem properly, which, in turn, makes the model more trustworthy and gives clinicians higher confidence in using it for medical diagnosis.

## 4. Conclusions

In this work, we explored a highly effective, end-to-end, deep neural network-based approach for the detection of fetal QRS complexes in non-invasive fetal ECG. We extended the novel octave convolution operation to time-series data of the non-invasive fECG, to address the temporal redundancy problem in conventional 1-D CNNs. The improvement in the computation and memory efficiency of the model facilitates its deployment on resource-constrained devices (e.g., wearables). Our proposed method achieved 91.82% of precision and 90.32% of recall on PhysioNet dataset while still demonstrating the robustness in practical scenarios with noisy data. Our approach holds promise to enable fetal and maternal well-being monitoring in the home setting, saving cost and labor as well as supporting the society in special pandemic scenarios such as the COVID-19. Moreover, with an approach to make the model more transparent by the interpretation of its decisions, the method would be better adopted by clinicians to augment diagnosis.

## Figures and Tables

**Figure 1 sensors-20-03757-f001:**
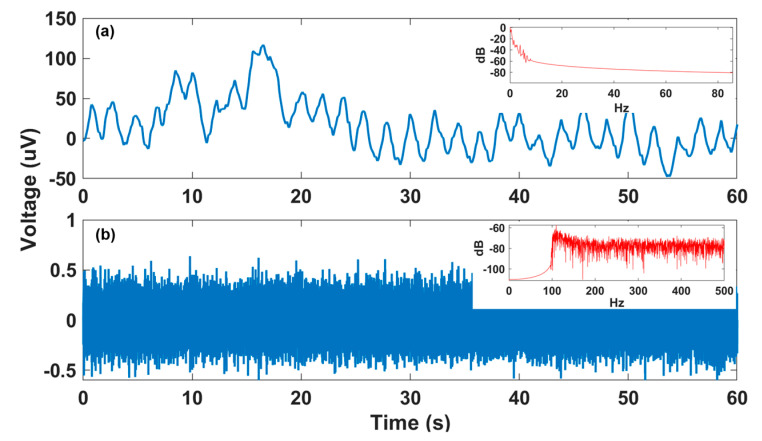
Decomposition of fetal/maternal ECG (f/mECG) into low-frequency and high-frequency components (from data a74-channel 1): (**a**) Low-frequency f/mECG part with the dominant frequencies of below 1 Hz, as shown in power spectral density (PSD) plot in the top right corner; (**b**) high-frequency f/mECG part dominantly belongs to the frequency of above 100 Hz, as shown in the PSD plot in the top right corner.

**Figure 2 sensors-20-03757-f002:**
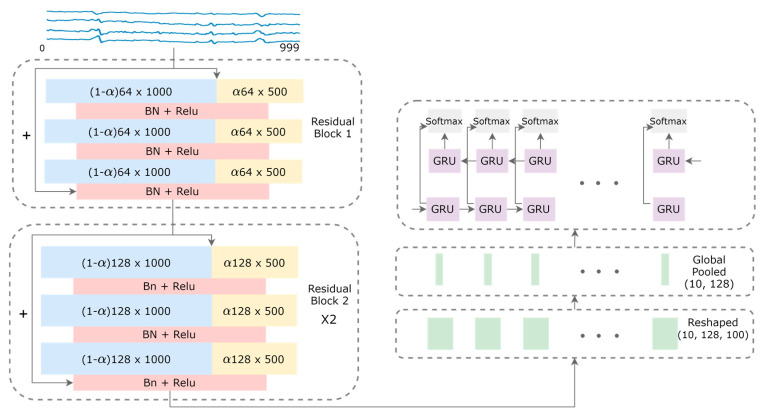
Model architecture for the fetal QRS complex detection. High-frequency and low-frequency feature maps are denoted by blue and yellow rectangles, respectively. The sizes of the feature maps are indicated inside the rectangles, with the hyper-parameter α that controls the ratio of channels allocated to the low-resolution features. Shortcut connections are denoted by the arrows with the plus signs.

**Figure 3 sensors-20-03757-f003:**
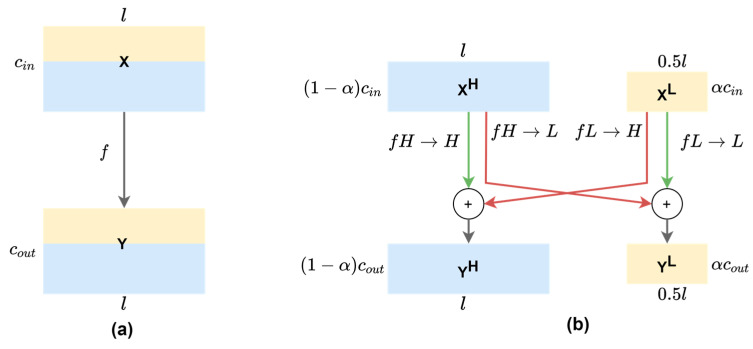
(**a**) 1-D vanilla convolution operation on mixed-frequency feature maps of the same resolution. (**b**) 1-D octave convolution on decomposed feature maps where low-frequency channels have 50% resolution. Red arrows denote inter-frequency information exchange $\left(f^{H\to L},f^{L\to H}\right)$, while green arrows denote intra-frequency information update $\left(f^{H\to H},f^{L\to L}\right)$, where *f^A^**^→B^* denotes the convolutional operation from feature map group A to group B. The *l* and *c* denote the temporal dimension and the number of channels, respectively. The ratio α of input channels (α_in_) and output channels (α_out_) are set at the same value throughout the network, except that the first OctConv has α_in_ = 0 and α_out_ = α, while the last OctConv has α_in_ = α and α_out_ = 0.

**Figure 4 sensors-20-03757-f004:**
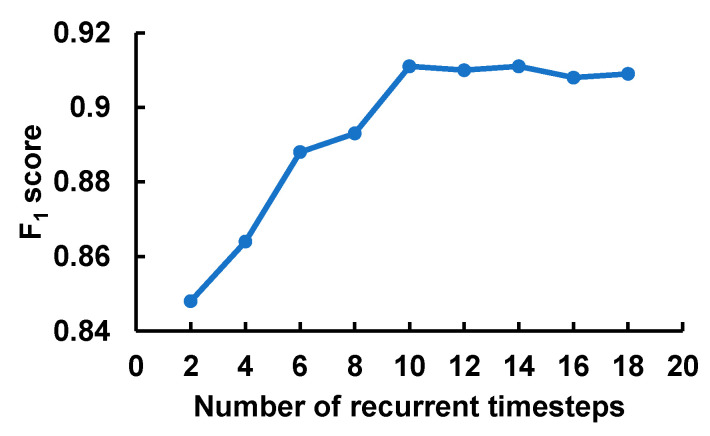
The impact of the number of recurrent timesteps that corresponds to the input sequence length.

**Figure 5 sensors-20-03757-f005:**
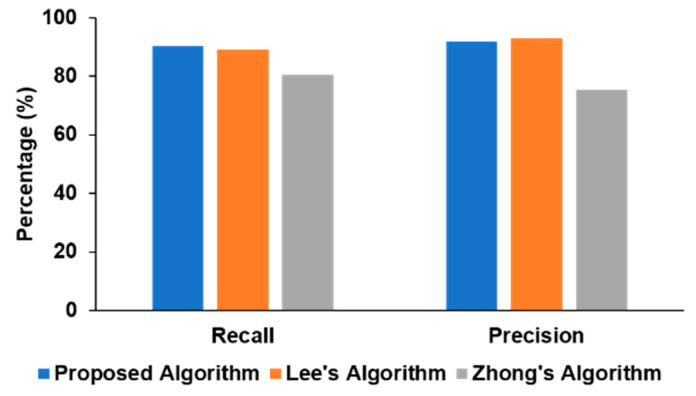
Performance comparison of different algorithms.

**Figure 6 sensors-20-03757-f006:**
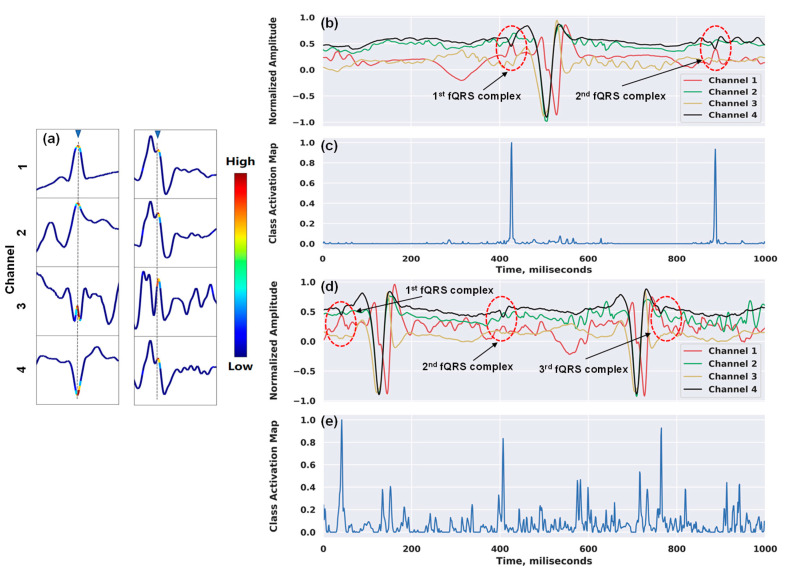
Examples of class activation maps for fetal QRS detections: (**a**) 4 × 100 window frames with annotated fQRS complex (marked in gray); (**b**) and (**d**) two input sequences of 4 × 1000 with two fQRS complexes and three fQRS complexes annotated, respectively; (**c**) two main spikes correspond to the fQRS complex positions of the signals in (**b**); (**e**) three main spikes correspond to the fQRS complex positions of the signals in (**d**).

**Table 1 sensors-20-03757-t001:** Theoretical gains of computational cost and memory consumption of 1-D OctConv (α = 0 is the case of vanilla convolution).

Ratio (α)	0	0.25	0.5	0.75
#FLOPs Cost	100%	78%	63%	53%
Memory Cost	100%	88%	75%	63%

**Table 2 sensors-20-03757-t002:** Performance on the original PhysioNet dataset. Inference time was measured on 420 4 × 1000 instances with Nvidia GeForce GTX 1060 Max-Q GPU (Nvidia, Santa Clara, CA, USA).

α	F_1_-Test	F_1_-Cross	CNN-GFLOPs	GRU-FC-GFLOPs	Inference Time (s)
0	0.907	0.872 ± 0.048	0.52	3 × 10^−4^	0.59
0.25	0.911	0.874 ± 0.054	0.42	3 × 10^−4^	0.55
0.5	0.901	0.869 ± 0.059	0.34	3 × 10^−4^	0.48
0.75	0.894	0.866 ± 0.058	0.29	3 × 10^−4^	0.45

**Table 3 sensors-20-03757-t003:** F_1_ score on the noisy dataset with the model’s α of 0.25.

Types of Noise	SNR Level (dB)			Motion Noise
	50.6	36.8	29.12	
F_1_	0.815	0.739	0.627	0.844

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
