# Peer review of "An Efficient and Robust Deep Learning Method with 1-D Octave Convolution to Extract Fetal Electrocardiogram"

_sensors, 2020, doi:10.3390/s20133757_

Round 1
Reviewer 1 Report
The authors propose an deep neural network that detects fQRS complexes from multi-channel non-invasive fECG, without requiring a reference maternal ECG signal.
I have several remarks on the content:
1. The principle relies strongly on the octave convolution I was not capable to understand from the present article. I could of course have had a look on the reference but the architecture is recent (2019). Hence the authors should make more efforts and take time to eplain it. The figure 3 is not comprehensive even with the explanations.
2. May be part of the response to the 1st remark is the Figure 2 which is missing.
3. In Eq (10) (by the way, reference should be on the right not in the middle), I can read the binary cross entropy loss function but where come the \beta parameter from ?
4. line 277: you compare 90.32% of success with 89.0%. Statistically this is not a significantly different.
I am questionning myself on the usefulness of a detector of QRS complex; usually, a simple low pass filter is sufficient to "see" fetal QRS by cancelling the lowx frequency wave...
But OK. The Kalman filter may be better (even if I am not convinced) by it mlust be parametrized...
An other point: you (authors) check various DLNN (only two, in fact, that is not so much) but above all you didn't check the capability of your model to properly identify the QRS complex for various SNR, shape of the QRS,etc.
In summary, the conclusions are your work are at minus limited from the data and the model point of view and the results are not amazing.
In other words, concerning the data, grade your recordings by complexity, and then produce your results according to this graduation scale.
And please, use a reference model as...a reference to compare with...
I suggest the authors refund their experiments, they are too weak.
Typos:
- 37: "to to visit"
- 71: "employed by Lo at el. [11]"
- 119: "and then by using extended Kalman filter, motion noise and filtered ECG data are achieved."
badly said
- 130 and 149: same equation numbers !
uniformize mathematical notations through your manuscript:
e.g. eq. (8) and (9) the parameter name alpha is probably the same, or the X in eq(1) line 130 and the X in line 131
- the references to the figures in the text are written in bold. Which is not request by the editor.
Reviewer 2 Report
The paper presents an Octave Convolution-based method for fetal QRS complex detection. I consider that paper methodologically sound and I have simply some technique questions or request of requirements:
1. I have not found the Figure 2 in the manuscript.
2.P3, line 113. The paper should offer more details about the amplitude range of noise levels 3, 6, and 9, or use SNR(signal to noise ratio) to measure the noise levels.
3. P7, line 238. ϵ = 1e−8. What is ϵ?
4. For the purpose of illustration, it is recommended that the annotations of the locations of fetal QRS should be marked in Figure 6.
Reviewer 3 Report
This paper presents a fetal QRS detector from four-lead maternal abdominal electrocardiograms using a one-dimensional version of the multiresolutional convolution network OctConv and Gated Recurrent Units. The training and test data come from 75 one-minute recordings of the 2013 Physionet/CinC Challenge (setA).
In general the description of the approach is clear and the technical basis is sound.
The deep-learning approach is not well situated within classical signal processing techniques. You should include the reference
Kahankova R, Martinek R, Jaros R, et al. A Review of Signal Processing Techniques for Non-Invasive Fetal Electrocardiography. IEEE Rev Biomed Eng. 2020;13:51,73. doi:10.1109/RBME.2019.2938061
Please perform cross-validation to obtain means and variances of your performance over the folds with this dataset, to better support the generalizability of the approach.
Please situate your performance in the context of other studies.
Fig 4. Three samples on the plateau does not seem to be sufficient evidence to conclude that the F1 vs. recurrent timesteps performance has "levelled off".
Fig. 6. The caption should be sufficiently descriptive to be understood on its own. Please amplify the description "class activation heatmaps" (this phrase does not occur in the text either). Furthermore, if the GRAD-CAM technique is meant to be a more explanatory interpretation of the detection, it should be understandable by both technical and clinical users.
There are numerous manuscript problems that could have been avoided with careful proofreading.
Figure 2 seems to be missing. Please add an overall architectural diagram in any case.
Equation 10: please distinguish targets y and estimates y_hat.
Table 2 is split between pages.
There are numerous incomplete references (e.g. 12, 13, 14).
Typos and grammar:
"Alternatively, fetal heart monitoring can also be done via fetal electrocardiogram" (fECG) -->
"Alternatively, fetal heart monitoring can also be done via fetal electrocardiogram (fECG) acquired from the fetal scalp".
Also this is where you should clarify the problem of possible infection (this is not clear when reading the abstract).
awkward: is presented as 1000 samples per sample per second ->
each channel acquired at 1000 Hz.
we deploy the concept Bi-GRUs->
we deploy Bi-GRUs
In OctConv, low-frequency tensor is stored->
In OctConv, a low-frequency tensor is stored
Table 1
".0" -> 0
awkward: "Besides, the F1 score on the dataset..."
Round 2
Reviewer 3 Report
Most of the issues have been addressed by the authors.
However, there is still no y_hat in the binary cross-entropy equation (now equation 11). Distinguish which of the y_ij are ground truth y and which are predictions y_hat.
Author Response
- However, there is still no y_hat in the binary cross-entropy equation (now equation 11). Distinguish which of the y_ij are ground truth y and which are predictions y_hat.
Thank you for pointing that out! There is actually y_hat in the original Word file (y*log y_hat - (1-y)*log(1-y_hat)). It was missing due to some glitches in the system while converting to the PDF file. We also notice the editorial office about this issue.